# A Biodegradable Tissue Adhesive for Post-Extraction Alveolar Bone Regeneration under Ongoing Anticoagulation—A Microstructural Volumetric Analysis in a Rodent Model

**DOI:** 10.3390/ijms25084210

**Published:** 2024-04-10

**Authors:** Marius Heitzer, Philipp Winnand, Mark Ooms, Zuzanna Magnuska, Fabian Kiessling, Eva Miriam Buhl, Frank Hölzle, Ali Modabber

**Affiliations:** 1Department of Oral and Maxillofacial Surgery, University Hospital RWTH Aachen, Pauwelstraße 30, 52074 Aachen, Germanymooms@ukaachen.de (M.O.);; 2Institute for Experimental Molecular Imaging, RWTH Aachen University, Forckenbeckstraße 55, 52074 Aachen, Germany; zmagnuska@ukaachen.de (Z.M.);; 3Institute for Pathology, Electron Microscopy Facility, University Hospital RWTH Aachen, Pauwelstraße 30, 52074 Aachen, Germany

**Keywords:** bone regeneration, tissue adhesives, biodegradable adhesive, tooth extraction, socket preservation

## Abstract

In addition to post-extraction bleeding, pronounced alveolar bone resorption is a very common complication after tooth extraction in patients undergoing anticoagulation therapy. The novel, biodegenerative, polyurethane adhesive VIVO has shown a positive effect on soft tissue regeneration and hemostasis. However, the regenerative potential of VIVO in terms of bone regeneration has not yet been explored. The present rodent study compared the post-extraction bone healing of a collagen sponge (COSP) and VIVO in the context of ongoing anticoagulation therapy. According to a split-mouth design, a total of 178 extraction sockets were generated under rivaroxaban treatment, of which 89 extraction sockets were treated with VIVO and 89 with COSP. Post-extraction bone analysis was conducted via in vivo micro-computed tomography (µCT), scanning electron microscopy (SEM), and energy-dispersive X-ray spectroscopy (EDX) after 5, 10, and 90 days. During the observation time of 90 days, µCT analysis revealed that VIVO and COSP led to significant increases in both bone volume and bone density (*p* ≤ 0.001). SEM images of the extraction sockets treated with either VIVO or COSP showed bone regeneration in the form of lamellar bone mass. Ratios of Ca/C and Ca/P observed via EDX indicated newly formed bone matrixes in both treatments after 90 days. There were no statistical differences between treatment with VIVO or COSP. The hemostatic agents VIVO and COSP were both able to prevent pronounced bone loss, and both demonstrated a strong positive influence on the bone regeneration of the alveolar ridge post-extraction.

## 1. Introduction

Complications commonly encountered after tooth extraction in patients undergoing anticoagulation therapy include post-extraction bleeding and alveolar bone resorption. Sufficient hemostasis is of eminent importance as it is the basis for the complex wound healing processes of an extraction socket [1]. Tooth extraction is the trigger for biological events that cause remodeling of the extraction socket and lead to the resorption of the alveolar ridge [1,2,3]. Existing strategies aim to minimize surgical trauma, stabilize the coagulum inside the wound area, and develop a dynamic osteogenetic microenvironment. Minimally invasive procedures include powered periotomes, piezosurgery, physics forceps, and vertical extrusion systems, and compared to conventional extraction procedures they decrease the risk of unfavorable fracture of the facial bone plate [4]. Agents such as oxidized cellulose, collagen sponges [1], fibrin glue, cyanoacrylate adhesive, and bone glues [5] aim to stabilize the coagulum, whereas platelet-rich plasma gel and topical thrombin are intended to improve the osteogenic microenvironment [6,7]. However, none of these procedures or agents have yet been implemented successfully in clinical application, which is why there is a great need for novel approaches.

The bone defect repair of extraction sockets remains a tremendous challenge for clinicians, and focus is increasingly being placed on preserving the existing alveolar ridge to reduce resorption, which is of crucial importance for implant placement and many prosthetic restorations [1,5,8,9,10,11]. Inadequate bone conditions impair the feasibility of implant placement and increase the risk of implant failure [9,10,11]. For adequate bone defect repair, the main goal is to minimize external bone resorption and to support internal bone formation within the extraction socket. Techniques of ridge preservation often involve the application to the extraction sockets of bone graft materials or biomaterial scaffolds. However, because allogeneic and xenogeneic bone substitutes trigger different inflammatory antigen-driven reactions, the focus of research has increasingly been on the search for other biomaterials from which to construct scaffolds [1,9]. Nevertheless, no transplantation method has yet emerged as the ideal procedure for the preservation of socket bone because its use is associated with considerable limitations, especially when ancillary procedures, such as hemostatic agents, are required.

Minimally invasive procedures are increasingly being used in tooth extraction in order to enhance the regenerative potential of the extraction socket and to reduce postoperative complications [12,13].

Collagen is a protein that is a physiologically significant component of the extracellular matrix and is considered a suitable biomaterial for hemostasis and the preservation of the alveolar bone after tooth extraction [1,2,6,14,15,16]. Therefore, collagen scaffolds are increasingly included in current research to positively influence the bone remodeling of the extraction socket [1,2,16]. On the other hand, collagens have demonstrated considerable disadvantages in the extraction sockets of anticoagulated patients due to their poor enzyme resistance [1] and lack of hemostyptic effects [17], which is the reason why alternative methods are urgently needed for post-extraction alveolar bone regeneration and hemostasis in patients undergoing anticoagulation therapy.

Tissue adhesives are the subject of numerous scientific research questions [18,19], and are often used for their beneficial hemostyptic effects [14,20,21]. The utilization of various tissue adhesives after tooth extraction is not a new approach [5,14,22,23,24,25]. For example, fibrin glue [24,25], cyanoacrylates [22,23], and polyvinyl alcohols [5] have been used in extraction sockets with the intention of facilitating surgical procedures [23] and improving the wound healing of bony tissue [5,23,24]. Although tissue adhesives have demonstrated promising results in terms of a positive influence on bone metabolism and the prevention of increased bone resorption post-extraction [5,22,23,24], their use within an extraction socket has thus far been limited exclusively to animal models [5,23,24]. To date, no adhesive has been successfully established for clinical use inside of extraction sockets.

In our recently published article, the novel tissue adhesive VIVO (Adhesys Medical GmbH, Aachen, Germany) was successfully used over a period of five days in a rodent study on hemostasis after extraction under oral anticoagulation (OAC) therapy [14]. In addition to the favorable hemostyptic properties in various bleeding test modalities [14,20,21,26], a favorable tissue compatibility, as evidenced by good biocompatibility during the various degradation phases in soft tissues, has already been demonstrated [26]. Post-extraction bleeding and bone resorption of the alveolar bone are two complications that lead to significant limitations in the healing phase and compromised bone sites [1]. Therefore, the use of VIVO in the extraction sockets of patients undergoing anticoagulation has the potential to create favorable conditions for bone regeneration, as well as a reduction in alveolar bone resorption compared to existing procedures, and reliable hemostasis.

This in vivo study primarily aimed to evaluate the effect of two different hemostyptic materials on post-extraction bone regeneration in anticoagulated rats with micro-computed tomography (μCT) images, energy-dispersive X-ray spectroscopy (EDX), and scanning electron microscopy (SEM). In this rodent split-mouth study, VIVO was compared to collagen sponges (COSP), which were both applied directly after tooth extraction and were observed post-operatively over a period of 90 days.

The null hypothesis of the present study was that the use of the novel polyurethane adhesive would be equally effective and show no differences in bone resorption compared to the use of the gold standard COSP after tooth removal under rivaroxaban therapy.

## 2. Results

In total, 89 animals with 178 extraction sides of the first upper molar were included in this study. Only clinically irrelevant oozing, which required no further therapy, was observed to the same extent in COSP and VIVO with ongoing rivaroxaban administration.

The surgical sides of the upper jaw and the analyzed jaw segments from µCT scans are shown as 3D reconstructions in Figure 1 and Figure 2.

The analyses of the µCT scans (BV/TV, BMD, Tb.Th, Tb.Sp, ΔBV/TV, ΔBMD, ΔTb.Th, and ΔTb.Sp) after 5 days, 10 days, and 90 days post-extraction are shown in Table 1 and Figure 3 and Figure 4. After 5 days, minor bone resorption could be observed. The ΔBV/TV of the extraction sockets of the COSP-treated group was −0.21 ± 0.16%. Similarly, the ΔBV/TV of the VIVO-treated group was −0.14 ± 0.13%. In the course of remodeling, there was an increase in alveolar bone. The ΔBV/TV increased to 0.46 ± 0.29% (*p* ≤ 0.001) with the COSP treatment and 0.48 ± 0.2% (*p* ≤ 0.001) with the VIVO treatment after 90 days. A comparable increase was observed in both groups in terms of the differences in BMD. Analogous to ΔBV/TV, the extraction sockets treated with COSP had a decrease in BMD of −0.1 ± 0.06 g/cm^3^ after 5 days, which significantly increased to a higher density of mineral bone 0.11 ± 0.05 g/cm^3^ (*p* ≤ 0.001) after 90 days.

Analogously, in the VIVO-treated extraction sockets, the ΔBMD was −0.01 ± 0.08 g/cm^3^ after 5 days and 0.09 ± 0.04 g/cm^3^ (*p* ≤ 0.001) after 90 days. In addition, the analysis of ΔTb.Th showed a significant increase in both the COSP- and VIVO-treated groups between 5 and 90 days (both *p* ≤ 0.001). No statistical differences in ΔTb.Sp were found at any observation time point in the bone of the COSP-treated sockets and the VIVO-treated sockets. Furthermore, no statistically significant differences were found between the COSP and VIVO treatment groups in terms of ΔBMD, ΔTb.Th, or ΔTb.Sp at 5 days, 10 days, or 90 days.

SEM revealed that five days after molar extraction, both treatment procedures resulted in thin new bone formation, which was detected as lamellar bone structures with formative osteocytic lacunae and distinct bone-forming areas of the alveolar bone surfaces. After ten days, the extent of new bone formation in the VIVO- and COSP-treated extraction sockets had obviously increased further. Bone resorption was also observed in the areas closely adjacent to the bone-forming areas. After 90 days, both the VIVO-treated and COSP-treated extraction sockets were similarly and predominantly filled with lamellar bone mass. Apart from a few absorption gaps, most of the alveolar bone surfaces were covered with newly formed lamellar bone layers (Figure 5a).

For the EDX analysis, two samples for each treatment after 5 days, 10 days, and 90 days were used for the evaluation (Table 2 and Figure 5b). Over the observation time of 90 days, a non-significant increase in the weight percentages of calcium, carbon, oxygen, and phosphate content to the organic component was observed in both treatment groups, which reflected the degree of rebuilt bone matrix. The quantitative analyses of the calcium-to-carbon ratio (Ca/C) and the calcium-to-phosphate ratio (Ca/P) within the extracted alveolar sockets revealed that both the Ca/C and Ca/P indicated newly formed bone matrixes with both treatments. The bone composition at the mineral level of the COSP-treated alveoli was similar to those treated with VIVO, with no significant difference at 5, 10, or 90 days after molar extraction.

## 3. Discussion

Interventions that can simultaneously resolve both post-extraction hemorrhage and alveolar bone resorption are in high demand. Hemostasis is extremely important for the healing of an extraction socket [1], especially because clinical and experimental studies have described a reduction in the original ridge width of up to 50% during the post-extraction socket healing process [3]. In the present study, a novel procedure based on the use of the polyurethane adhesive VIVO and its bone regenerative influence against the background of post-extraction bone remodeling was evaluated over a period of 90 days. In the literature, VIVO is described as quick to prepare, easy to handle, fast-curing, cost-effective, biocompatible, and with favorable biodegradation [26]. Especially, the biodegradability of VIVO after subcutaneous implantation in soft tissue has already been documented over a period of 2 years. Bremer et al. [26] describe a complete degradation of the adhesive within the tissue as early as 6 months after implantation, as well as good biocompatibility during the various degradation phases. After the degradation of the adhesive, complete remodeling of the physiological tissue was observed [26]. This study is the first to describe the influence of the biodegradable adhesive VIVO on bony tissue and bone remodeling.

Hulsart-Billstrom et al. described a considerable need for in vitro test procedures that can investigate the benefits of regenerative materials so that the burden of animal testing can be reduced [27]. At present, however, the results of biomaterials obtained by in vitro studies cannot adequately reflect the complex healing processes; thus, in vivo studies remain necessary [27]. Therefore, animal models are still important to understand the complex pathophysiological orchestration of bone healing and post-operative bone remodeling after dental extraction. The use of an established animal model for the assessment of post-extraction healing offers the advantage of the comparison of examination parameters as well as potential transferability to humans. As frequently described in the literature [1,14,23,24,28], the extraction of the first upper molar was performed to subsequently evaluate the bony healing processes of the extraction socket under the influence of COSP or VIVO in a living organism. Nevertheless, the used rat model must be considered a limitation, as biological discrepancies in orofacial bone regeneration, including the healing of tooth extraction sockets, between rodents and humans are described in the literature [29]. Considering that no animal model perfectly matches the human situation, this limitation in the transferability of the observed results to humans must be critically considered.

The literature frequently describes pronounced bone remodeling processes, which can sometimes be accompanied by considerable bone loss after tooth extraction [1,2,3,10,11,30]. Therefore, osteoinductive potential is among the most important requirements for a material that needs to ensure the favorable bone healing of the alveolar ridge after extraction, which manifests itself in the formation of newly formed bone within the extraction socket [1,11,15,16,23,24]. In our investigation, initial decent bone loss was observed in the COSP- and VIVO-treated groups with regard to BV/TV and BMD within the first ten days after extraction. Maia et al. [23] described how the use of adhesives in the extraction sockets of the maxillary molars of rats can lead to a few days’ delay in bony regeneration in the sockets. After a few days, this circumstance equalized, and no differences in the strength of the bone regeneration were observed despite the initial adhesive-associated delay [23]. Consistent with the descriptions in the literature, both COSP and VIVO exhibited positive bone regeneration and new bone formation after ten days. During the following observation time of 90 days, BV/TV significantly increased in the COSP-treated (*p* ≤ 0.001) and VIVO-treated (*p* ≤ 0.001) extraction sockets at 0.46% and 0.48%, respectively. 

In line with our findings, Gu et al. demonstrated a BV/TV of approximately 0.41% in the extraction sockets of the first upper molars in anticoagulated rats treated with collagen inlays after seven days [1]. Moreover, their study showed that the control group without treatment had a BV/TV of approximately 0.32%. One limitation of our study was the lack of a control group without therapy. Instead, the influence of VIVO was evaluated against the gold standard treatment of hemostyptic measures after tooth extraction under ongoing anticoagulation. Therefore, future studies should include a control group because a control group can more accurately discriminate the strength of the influence of VIVO versus COSP.

Post-extraction bleeding can result in poor healing results, which is reflected in increased bone resorption in the course of bone remodeling [1]. Therefore, in addition to osteogenic potential, the formation of a stable coagulum within the extraction socket is another important requirement of a suitable bone regenerative material. A stable coagulum is the prerequisite for the migration of inflammatory cells, which initiate the healing of the extraction socket through the formation of granulation tissue [30]. The use of VIVO to improve hemostasis has proven to be a favorable hemostyptic in various animal models with bleeding [14,20,21,26]. In this study, the hemostyptic effects of VIVO played a subordinate role as we were already able to demonstrate the hemostyptic effectiveness of VIVO post-extraction over a period of five days in a recently published pilot study [14]. For this reason, clinical investigations into post-operative bleeding with regard to the known hemostyptic properties of VIVO were not part of this study. Instead, a comparison of two hemostyptic preparations was conducted within the context of the effect of coagulation stabilization and the associated influence on bone remodeling after tooth extraction under ongoing OAC.

In the literature, collagens are often described as having weak hemostatic properties and disadvantages with regard to bone healing in the extraction sockets of anticoagulated patients [1,16,17]. In this study, this disadvantage could not be demonstrated, and the extraction sockets treated with COSP showed comparable bone regeneration in terms of BV/TV, BMD, Tb.Ts, and Tb.Sp compared to VIVO therapy over the 90-day study period.

In addition to osteoconductivity, osteoinductivity is an important material property for bone regeneration [31]. The addition of substances that increase the osteoinductivity of a material by preventing premature material degradation or influencing mechanical stability offer the potential to improve the osteogenic regeneration of biomaterials when used in extraction sockets [1,5,16,24]. The osteoinductivity of the polyurethane adhesive can be improved by the addition of human bone morphogenetic proteins or extracellular vesicles. It is therefore necessary that future studies investigate potential bioactive additions that can promote the bony regeneration and osseoinductive properties of VIVO.

Histological methods for the morphological analysis of bones have the considerable disadvantage that they are labor-intensive and time-consuming. These methods have the additional disadvantages that samples can be damaged during histological processing and that repeated measurements of the same sample at different times are not possible [32]. Because of these limitations, various 3D visualization techniques are increasingly being used. The high consistency and accuracy of µCT measurements have established µCT examination as the procedure for the assessment of mineral bony structures in several studies, and it is therefore particularly suitable for the in vivo monitoring of bony regeneration [1,5,24,32]. Moreover, assessment with µCT is an established examination modality for microtomography analyses of bone volume fraction (BV/TV), bone density (BMD), trabecular thickness (Tb.Th), and the trabecular separation (Tb.Sp) of specimens [28,32]. In addition, comparisons between two-dimensional histology and three-dimensional µCT with regard to BV/TV, BS/TV, Tb.Th, and Tb.Sp have shown that the non-destructive, fast, and precise radiological analysis of small bone samples makes it possible to measure bone structures even without biopsies [33].

Examinations of bone composition at the mineral level can generate important insights into the condition of the bone [34,35]. For example, a reduced Ca/P ratio is associated with induced bone loss [35], while the Ca/C ratio can reflect the degree of calcification of the bone matrix at different stages [34]. The results of this study indicated that both the Ca/P and Ca/C ratios showed a non-significant tendency, indicating bone remodeling via positive bone regeneration. According to the examination parameters of bone morphology in the µCT, no difference could be observed between the VIVO and COSP treatments with regard to the ionic composition of the alveolar bone healing in this study using EDX analysis.

## 4. Materials and Methods

### 4.1. Ethics Statement

All of the experiments were conducted in accordance with the Animal Research: Reporting of In Vivo Experiments (ARRIVE) guidelines [36], with the EU directive of the European Parliament and of the Council on the protection of animals used for scientific purposes (2010/63/EU), and with the German animal protection law (Tierschutzgesetz, TSchG). Approval of the animal protocol was granted on 18 February 2021 by the Governmental Animal Care and Use Committee of the State of North Rhine-Westphalia under approved ID: 81-02.04.2020.A166; Landesamt für Natur-, Umwelt- und Verbraucherschutz Recklinghausen, Nordrhein-Westfalen, Germany.

### 4.2. Experimental Animals

Ninety-three male Sprague Dawley rats that were seven weeks of age and weighed 389.93 ± 28.31 g (Janvier Labs, Le Genest-Saint-Isle, France) were included in this study. This manuscript contains the details of this study’s investigations and the radiological results. All rats were housed under a 12 h light/12 h dark cycle. The rats were provided with food and water ad libitum, with only soft-soaked food administered for seven days after dental extraction. Three rats per cage were kept in a pathogen-free environment in filter-top cages (Type 2000, Tecniplast, Buguggiate, Italy) with low-dust wood granulate bedding (Rettenmeier Holding AG, Wilburgstetten, Germany) and nesting material (Nestlet, 14010, Plexx B.V., Elst, The Netherlands). Systemic medication with a therapeutic dose of rivaroxaban 3 mg/kg was administered parenterally for ten days according to an established protocol [14]. The first administration was performed 15 min before the surgical procedure, and rivaroxaban injections were repeated daily over a period of ten days at the same time each day.

### 4.3. Surgical Procedures

An established surgical tooth extraction model was used [1,14,23,24,28] to evaluate the effects of COSP and VIVO on post-extraction bleeding and socket healing under ongoing rivaroxaban treatment. All operations were executed by a single and experienced surgeon. Under general anesthesia with an intraperitoneal injection of a combination of medetomidine (0.25 mg/kg) and ketamine (80 mg/kg), rivaroxaban concentrations were determined immediately prior to surgery according to an established protocol [14]. The rivaroxaban concentrations measured at the time of surgery were 266.7 ± 209.2 ng/mL. After a local submucosal injection of Ultracaine 4%, the gingiva around the right maxillary first molar was carefully separated with a dental probe. The extraction and osteotomy of the first maxillary molars were performed on both sides under microscopic magnification (OPMI pico f170, Carl Zeiss AG, Oberkochen, Germany). During the extractions, no bone fractures occurred, and in the rare case of root fractures these could be removed without any complications.

After the extraction and osteotomy of the tooth, the left socket was filled with VIVO and the right socket was filled with a colloid gelatin sponge (COSP; ROEKO Gelatamp forte, Coltene, Altstätten, Switzerland) according to a split-mouth design. After the development of a mucoperistal flap, the wound margins were adapted with single-button sutures (Vicryl 6-0, Ethicon Inc., Somerville, NJ, USA) (Figure 6). In the event of post-operative bleeding, the insertion of gauze was performed under slight pressure. In total, four animals died during the operation under general anesthesia without bleeding. The animals were divided into the following groups according to the study periods: the 5-day group (*n* = 29), the 10-day group (*n* = 31), and the 90-day group (*n* = 29).

### 4.4. Micro-Computed Tomography (µCT) Analysis

The µCT system (U-CT OI, MILabs, Utrecht, The Netherlands) was used for in vivo imaging under general anesthesia using isoflurane (induction with 5 vol% isoflurane  +  5 L O_2_/min; maintenance with 2 vol% isoflurane  +  2 L O_2_/min) (Abbott GmbH & Co. KG, Wiesbaden, Germany) imaged immediately post-operatively (T1) and after 5 days (T2), after 10 days (T3), and after 90 days (T4). The µCT imaging was performed with a voltage of 65 kV, power 0.13 mA, and 300 ms exposure time. The scans were acquired using ultra-focus magnification with 360° rotation at 0.75° increments with 0.3 s/degree, and the data were reconstructed at a 40 µm isotropic voxel size. For analysis, as previously described, the region of interest (ROI) (comprising the upper jaw, including the operation site) was selected and cut out of the original µCT [37] (Figure 2).

This way of analysis allowed easier handling and, in the meantime, kept the original high resolution of the acquired in vivo scan. For the analysis of bone healing, the treated extraction sockets and surrounding bone tissue were semi-automatically segmented in micro-CT images using a threshold of 1600 HU. The obtained segments were split into left (VIVO) and right (COSP) (Figure 3). This procedure was reproduced for each subject and for all time points. Group affiliation was not known during the analysis. Subsequently, the segmented regions of interest were analyzed in terms of bone volume fraction (BV/TV), bone mineral density (BMD), trabecular thickness (Tb.Th), and separation (Tb.Sp). The extracted parameters were statistically compared between VIVO and COSP and between the different time points after 5 days, after 10 days, and after 90 days.

### 4.5. Scanning Electron Microscopy (SEM) and Energy-Dispersive X-ray Spectroscopy (EDX) Analysis

Directly after sacrificing the animals after 5 days, 10 days, or 90 days, the SEM and EDX analyses were conducted for each jaw side (COSP or VIVO) of the upper rat jaw. The region of interest for sample collection was defined as the midpoint of the extraction socket. For each method, nine collected samples were used. The samples for SEM were fixed in 3% glutaraldehyde in 0.1M Sorensen’s phosphate buffer, dehydrated in an ascending ethanol series (30–100%), and dried at 37 °C (SEM and EDX) according to a previously published study [25]. The samples were analyzed using an environmental scanning electron microscope (ESEM XL 30 FEG; FEI, Eindhoven, The Netherlands) in backscatter mode with an acceleration voltage of 15 kV.

EDX analysis was performed with the EDAX Genesis system (EDAX, Mahwah, NJ, USA). As previously described, EDX analyses were performed at three random measurement points for each sample image using a mean value for statistical analysis. EDX analysis measured elements of bone composition, such as carbon, oxygen, natrium, phosphate, sulfur, and calcium [34].

### 4.6. Statistical Analysis

Analyses were performed using GraphPad Prism 10.1.1 (GraphPad Software, Inc., La Jolla, San Diego, CA, USA). The Kruskal–Wallis test was used for nonparametric independent variables to compare the differences between the parameters after a Gaussian normal distribution was excluded in the Shapiro–Wilk test. All data are represented as mean ± standard deviation (SD). Differences were considered significant when *p* ≤ 0.05. Post hoc power analysis was performed using F-test ANOVA to determine the power of 99% (G*Power, Version 3.1.9, Düsseldorf, Germany; Faul et al. [38,39]), based on the total sample size and a number of two treatments (COSP and VIVO), for an effect size of 0.365 and an α of 0.05.

## 5. Conclusions

This study demonstrated that the use of the polyurethane-based biodegradable tissue adhesive VIVO could generate promising results in promoting alveolar regeneration post-extraction under anticoagulation. However, against the limiting background of a rodent study, the results also showed that the gold standard COSP of local haemostyptic measurement in the extraction sockets of anticoagulated individuals had exactly the same positive effect on alveolar ossification as VIVO. Therefore, further studies are required to optimize the osteoinductive potential of VIVO and to further improve the influence of VIVO on bone regeneration in extraction sockets.

## Figures and Tables

**Figure 1 ijms-25-04210-f001:**
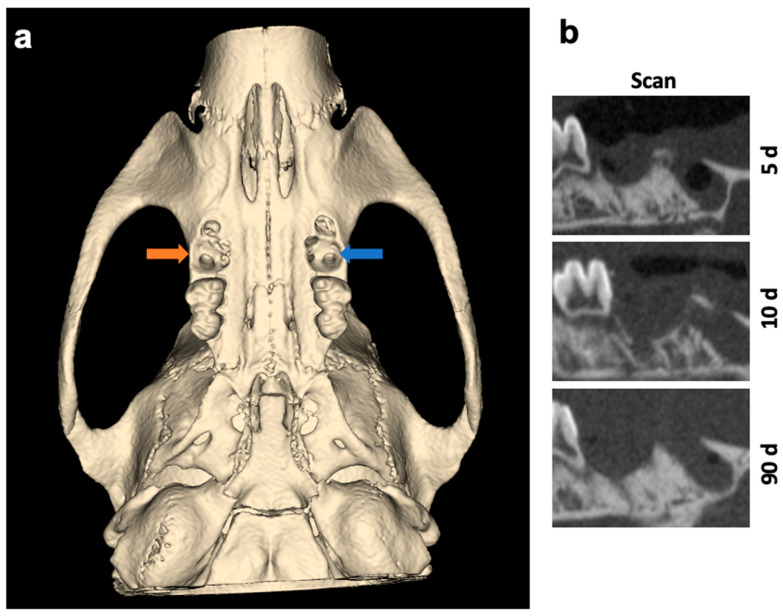
(**a**) Three-dimensional reconstruction and overview of the surgical sides of the upper jaw. Left socket (blue arrow) was treated with VIVO and the right socket (orange arrow) was treated with COSP. (**b**) Sagittal 2D images of the extraction sockets after 5, 10, and 90 days. Abbreviations: d = days.

**Figure 2 ijms-25-04210-f002:**
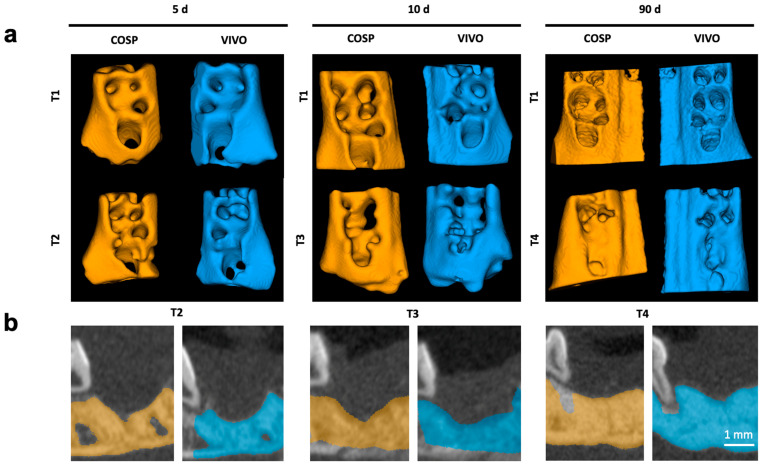
(**a**) Three-dimensional reconstruction of upper jaw segments with comparison before and after 5, 10, and 90 days. (**b**) Color highlighting of the bony parts of the maxilla in the sagittal plane; Segments of extraction sockets treated with VIVO (blue jaws) and treated with COSP (orange jaws). Abbreviations: d = days, T1 = imaged immediately post-operation, T2 = imaged after 5 days, T3 = imaged after 10 days, and T4 = imaged after 90 days.

**Figure 3 ijms-25-04210-f003:**
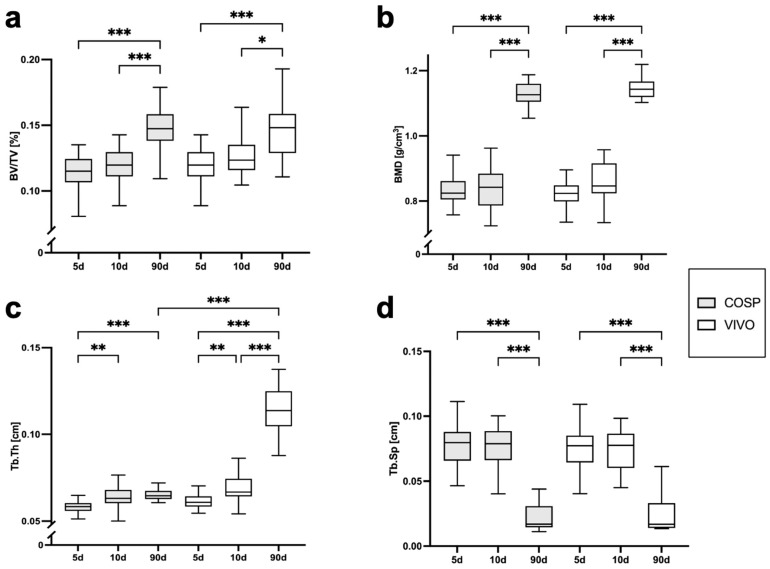
Graphical representation of (**a**) BV/TV, (**b**) BMD, (**c**) Tb.Th, and (**d**) Tb.Sp of VIVO and COSP treatment post-extraction after 5, 10, and 90 days. The data were analyzed using the Kruskal–Wallis Test. * *p* ≤ 0.05, ** *p* ≤ 0.005, and *** *p* ≤ 0.001. Abbreviations: BV/TV = Bone volume fraction, BMD = bone mineral density, Tb.Th = trabecular thickness, Tb.Sp = trabecular separation, d = days, VIVO = polyurethan adhesive, and COSP = collagen sponge.

**Figure 4 ijms-25-04210-f004:**
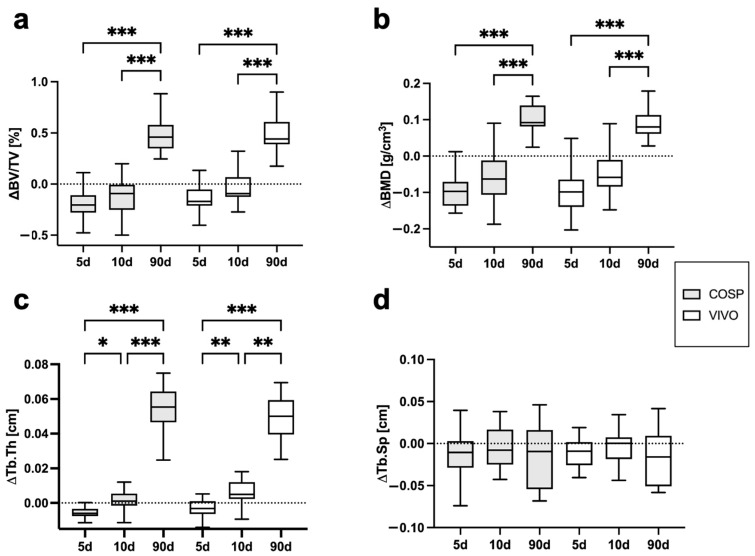
Graphical representation of (**a**) ΔBV/TV, (**b**) ΔBMD, (**c**) ΔTb.Th, and (**d**) ΔTb.Sp of VIVO and COSP treatment post-extraction after 5, 10, and 90 days. The data were analyzed using the Kruskal–Wallis test. * *p* ≤ 0.05, ** *p* ≤ 0.005, and *** *p* ≤ 0.001. Abbreviations: BV/TV = bone volume fraction, BMD = bone mineral density, Tb.Th = trabecular thickness, Tb.Sp = trabecular separation, d = days, VIVO = polyurethan adhesive, and COSP = collagen sponge.

**Figure 5 ijms-25-04210-f005:**
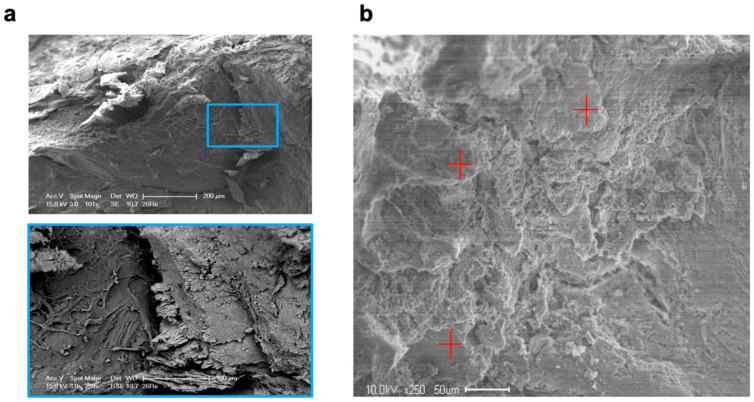
(**a**) SEM images and enlargement (blue box) of analyzed bone of an extraction socket treated with VIVO after 90 days. (**b**) SEM image in backscatter mode and EDX analyses at 3 random measurement points (red plus signs). Abbreviations: SEM = Scanning electron microscopy, EDX = Energy-dispersive X-ray spectroscopy.

**Figure 6 ijms-25-04210-f006:**
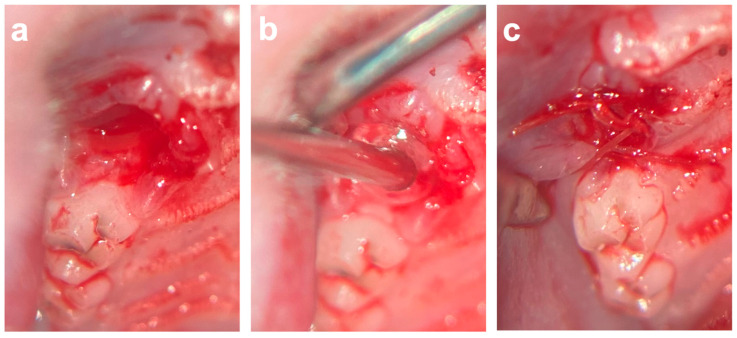
(**a**) Extraction socket of first upper molar. (**b**) Application of VIVO inside of the extraction socket. (**c**) After development of mucoperistal flap, the wound margins were adapted with single-button sutures. Abbreviations: VIVO = polyurethan adhesive.

**Table 1 ijms-25-04210-t001:** Numerical results from statistical analysis of µCT evaluation of BV/TV, BMD, Tb.Th, and Tb.Sp. The changes in Δ BV/TV, ΔBMD, ΔTb.Th, and ΔTb.Sp were evaluated using radiology (Δ µCT: T2-T1 for 5 days, T3-T1 for 10 days, and T4-T1 for 90 days). The data were analyzed with the Kruskal–Wallis test and are presented as mean values ± standard deviation (SD). Abbreviations: BV/TV = bone volume fraction, BMD = bone mineral density, Tb.Th = trabecular thickness, Tb.Sp = trabecular separation, VIVO = polyurethan adhesive, and COSP = collagen sponge.

	5 Days (n = 29)		10 Days (n = 31)		90 Days (n = 29)	
Parameter	COSP	VIVO	COSP	VIVO	COSP	VIVO
BV/TV [%]	0.113 ± 0.016	0.12 ± 0.129	0.120 ± 0.013	0.127 ± 0.015	0.148 ± 0.017	0.146 ± 0.02
ΔBV/TV [%]	−0.207 ± 0.155	−0.144 ± 0.129	−0.112 ± 0.178	−0.04 ± 0.144	0.463 ± 0.293	0.479 ± 0.201
BMD [g/cm^3^]	0.836 ± 0.051	0.831 ± 0.063	0.814 ± 0.16	0.861 ± 0.06	1.129 ± 0.04	1.143 ± 0.035
ΔBMD [g/cm^3^]	−0.097 ± 0.064	−0.098 ± 0.075	−0.061 ± 0.064	−0.049 ± 0.061	0.108 ± 0.049	0.093 ± 0.041
Tb.Th [cm]	0.058 ± 0.033	0.062 ± 0.004	0.064 ± 0.005	0.069 ± 0.007	0.065 ± 0.004	0.113 ± 0.015
ΔTb.Th [cm]	−0.006 ± 0.034	−0.003 ± 0.005	0.001 ± 0.005	0.006 ± 0.007	0.053 ± 0.016	0.048 ± 0.015
Tb.Sp [cm]	0.075 ± 0.022	0.074 ± 0.018	0.075 ± 0.017	0.075 ± 0.019	0.025 ± 0.016	0.026 ± 0.017
ΔTb.Sp [cm]	−0.015 ± 0.033	−0.015 ± 0.032	−0.004 ± 0.024	−0.003 ± 0.026	−0.015 ± 0.037	−0.017 ± 0.032

**Table 2 ijms-25-04210-t002:** EDX analysis and quantitative analysis of calcium, carbon, oxygen, sodium, phosphate, and sulphur weight % and Ca/C and Ca/P ratios in newly formed bone matrixes within extracted alveolar sockets after COSP and VIVO treatment. All data were analyzed with Kruskal–Wallis tests and represented as means ± standard deviation (SD). Abbreviations: Ca = calcium, C = carbon, P = phosphate, NS = non-significant, VIVO = polyurethan adhesive, and COSP = collagen sponge.

	5 Days		10 Days		90 Days		
Parameter	COSP	VIVO	COSP	VIVO	COSP	VIVO	*p* Value
Weight (%)							
Calcium	4.59 ± 7.14	1.81 ± 2.72	10.36 ± 11.61	8.46 ± 6.58	20.21 ± 6.13	20.77 ± 15.9	NS
Carbon	13.45 ± 6.84	32.16 ± 19.95	13.97 ± 5.91	36.75 ± 19.73	35.24 ± 12.53	43.66 ± 8.52	NS
Oxygen	6.90 ± 4.00	5.72 ± 3.98	6.95 ± 6.83	12.58 ± 5.37	16.66 ± 5.75	17.15 ± 9.72	NS
Sodium	0.68 ± 0.47	0.78 ± 0.40	1.22 ± 0.46	1.95 ± 0.69	1.19 ± 0.66	0.96 ± 0.59	NS
Phosphate	4.40 ± 3.92	3.64 ± 1.86	8.04 ± 6.47	6.05 ± 3.97	12.62 ± 2.63	15.01 ± 6.09	NS
Sulphur	0.79 ± 0.68	0.32 ± 0.27	0.41 ± 0.42	0.20 ± 0.19	0.05 ± 0.11	0.28 ± 0.38	NS
Ratios							
Ca/C	0.34	0.06	0.74	0.23	0.57	0.48	
Ca/P	1.04	0.5	1.29	1.4	1.6	1.38	

## Data Availability

The raw data supporting the conclusions of this article will be made available by the authors on request.

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
