# Peer review of "A Biodegradable Tissue Adhesive for Post-Extraction Alveolar Bone Regeneration under Ongoing Anticoagulation—A Microstructural Volumetric Analysis in a Rodent Model"

_ijms, 2024, doi:10.3390/ijms25084210_

Round 1

Reviewer 1 Report

Comments and Suggestions for Authors

The current manuscript is an interesting experimental research article on the assessment of the polyurethane adhesive VIVO’s application in post-extraction alveolar bone regeneration. It appears to be overall well-structured and many relevant assays were performed, and it also appears to be quite novel, adding an important contribute to bone regeneration science. Hence, I only advise that some alterations be made before acceptance for publication:

- In the introduction section, more should be said about current strategies to achieve a minimization of surgical trauma, stabilization of the coagulum inside the wound area, and the development of a dynamic osteogenetic microenvironment, and their limitations, in order to better support the need for a novel platform;

- Each figure caption should include statistically relevant information, such as the applied statistical tests;

- Abbreviations and their full description should also be included in each Figure caption;

- Limitations of the applied methods should be mentioned and discussed, such as the relevance of the selected animal model, and in what extent might it correlate to what happens in humans;

- The authors mention that the rats used in the study weighted 390 g; in this a mean value? Since I find it hard to believe that all 93 rats had the exact same weight, the weight range should be mentioned, for transparency purposes;

- VIVO biocompatibility should be mentioned, as well as potential associated side-effects (if any have already been identified);

- The results also showed that the gold standard COSP and VIVO had similar positive effects; in light of this, authors should better support the novel system giving more attention to other aspects aside from performance, such as general cost and biocompatibility;

- Necessary future studies should be specified and described;

- An abbreviation list is missing and should be added.

Author Response

Letter to the reviewers

Dear Editors and Reviewers,

Re:”A biodegradable tissue adhesive for post-extraction alveolar bone regeneration under ongoing anticoagulation - a micro-structural volumetric analysis in a rodent model”.

Thank you for your effort and time to improve our paper. Your comments are valuable and help us to increase the impact of the publication. We will answer to all your comments and will describe the changes we have made. All changes in the manuscript are marked.

Referee(s)' Comments to Author:

Reviewer 1:

The current manuscript is an interesting experimental research article on the assessment of the polyurethane adhesive VIVO’s application in post-extraction alveolar bone regeneration. It appears to be overall well-structured and many relevant assays were performed, and it also appears to be quite novel, adding an important contribute to bone regeneration science. Hence, I only advise that some alterations be made before acceptance for publication:

- In the introduction section, more should be said about current strategies to achieve a minimization of surgical trauma, stabilization of the coagulum inside the wound area, and the development of a dynamic osteogenetic microenvironment, and their limitations, in order to better support the need for a novel platform;

Thank you very much for your valuable objections which help us to improve the quality of our manuscript. The following passage was added to the manuscript to support the need for novel forms of treatment:

“Minimally invasive procedures include powered periotomes, piezosurgery, physics forceps or vertical extrusion systems and compared to conventional extraction procedures they decrease the risk of unfavourable fracture of the facial bone plate [4]. Agents such as oxidized cellulose, collagen sponges [1], fibrin glue, cyanoacrylate adhesive or bone glues [5] aim to stabilize the coagulum, whereas platelet-rich plasma gel and topical thrombin are intended to improve the osteogenic microenvironment [6, 7]. However, none of these procedures or agents have yet been implemented successfully in clinical application, which is why there is a great need for novel approaches.”

- Each figure caption should include statistically relevant information, such as the applied statistical tests;

The authors would like to thank you for the careful review and valuable assessment of our manuscript. Applied statistical tests were added to the figure captions as requested.

- Abbreviations and their full description should also be included in each Figure caption;

The authors would like to thank you for your careful review. The abbreviations and their full description were added to the figure caption as requested.

- Limitations of the applied methods should be mentioned and discussed, such as the relevance of the selected animal model, and in what extent might it correlate to what happens in humans;

Thank you for your comments regarding the limitations of the methods and the relevance of the selected animal model, and in what extent might it correlate to what happens in humans. The following part was added to the discussion section of the manuscript:

“Nevertheless, the used rat model must be considered a limitation, as biological discrepancies in orofacial bone regeneration, including the healing of tooth extraction sockets, between rodents and humans are described in the literature [30]. Considering that no animal model perfectly matches the human situation, this limitation in the transferability of the observed results to humans must be critically considered.“

- The authors mention that the rats used in the study weighted 390 g; is this a mean value? Since I find it hard to believe that all 93 rats had the exact same weight, the weight range should be mentioned, for transparency purposes.

We would like to thank you for the opportunity to describe more clearly the matter you have raised and apologize that we did not elaborate this clearly enough in our manuscript. The following weight range of rats was added to the manuscript:

“389.93 ± 28.31 g”

- VIVO biocompatibility should be mentioned, as well as potential associated side-effects (if any have already been identified).

According to your suggestion, we have included the following passages regarding to the insights of biocompability of VIVO:

“The biodegradability of VIVO after subcutaneous implantation in soft tissue has already been documented over a period of 2 years. Bremer et al. [26] describe a complete deg-radation of the adhesive within the tissue as early as 6 months after implantation as well as good biocompatibility during the various degradation phases. After degradation of the adhesive, complete remodeling to physiological tissue was observed [26]. This study is the first to describe the influence of the biodegradable adhesive VIVO on bony tissue and bony remodeling.”

- The results also showed that the gold standard COSP and VIVO had similar positive effects; in light of this, authors should better support the novel system giving more attention to other aspects aside from performance, such as general cost and biocompatibility.

Thank you very much for the important hint to focus more on other aspects such as biocompatibility and general costs under the circumstances mentioned. Accordingly, the authors have included the following passages in the manuscript:

“In the literature, VIVO is described as quick to prepare, easy to handle, fast curing, cost effective, biocompatible and with favorable biodegradation [26]. The biodegradability of VIVO after subcutaneous implantation in soft tissue has already been documented over a period of 2 years. Bremer et al. [26] describe a complete deg-radation of the adhesive within the tissue as early as 6 months after implantation as well as good biocompatibility during the various degradation phases. After degradation of the adhesive, complete remodeling to physiological tissue was observed [26]. This study is the first to describe the influence of the biodegradable adhesive VIVO on bony tissue and bony remodeling.”

- Necessary future studies should be specified and described.

Thank you for this advice, which helps us to improve our manuscript. 

“The osteoinductivity of the polyurethane adhesive can be improved by the addition of human bone morphogenetic protein or extracellular vesicles. It is therefore necessary that future studies investigate potential bioactive additions that can promote the bony regeneration and osseoinductive properties of VIVO.”

- An abbreviation list is missing and should be added.

Thank you very much for this valuable advice. The following abbreviation list was added to the manuscript:

BMD

Bone mineral density

BV/TV              

Bone volume fraction

C

Carbon

Ca

Calcium

COSP

Collagen sponge

d

Days

EDX

Energy-dispersive X-ray spectroscopy

µCT

Micro computed tomography

P

Phosphate

SEM

Scanning electron microscopy

T

Time of radiological imaging

Tb.Sp

Trabecular separation

Tb.Th

Trabecular thickness

Kind regards,

Marius Heitzer and colleagues

Reviewer 2 Report

Comments and Suggestions for Authors

Dear authors,

Thank you for submitting your valuable work to the journal. The topic of your research is interesting, as post-extractional alveolar bone resorption is one of the most significant issues in dental medicine. Thus, viable solutions for ridge preservation for future implant placement are required in order to limit the need for oral surgery and bone augmentation, offering minimally-invasive techniques to a wider range of patients.

The paper is generally well structured and well written. Nevertheless, there are some changes I would suggest in order to increase its scientific clarity:

- please add a null hypothesis to the aim of your study

- were the extractions performed by the same researcher?

- did any extractions resulted in complications, such as alveolar bone or root fracture during the extraction?

- why wasn't a control group, no COSP or VIVO and no anticoagulant, included in the study's design?

- the limitations of the study should be more broadly discussed

- no commercial interests to the COSP or VIVO should be disclosed

We look forward to receiving the revised version of your manuscript.

Kind regards!

Comments on the Quality of English Language

Minor check-up

Author Response

Letter to the reviewers

Dear Editors and Reviewers,

Re:”A biodegradable tissue adhesive for post-extraction alveolar bone regeneration under ongoing anticoagulation - a micro-structural volumetric analysis in a rodent model”.

Thank you for your effort and time to improve our paper. Your comments are valuable and help us to increase the impact of the publication. We will answer to all your comments and will describe the changes we have made. All changes in the manuscript are marked.

Referee(s)' Comments to Author:

Reviewer 2:

Dear authors, thank you for submitting your valuable work to the journal. The topic of your research is interesting, as post-extractional alveolar bone resorption is one of the most significant issues in dental medicine. Thus, viable solutions for ridge preservation for future implant placement are required in order to limit the need for oral surgery and bone augmentation, offering minimally-invasive techniques to a wider range of patients.

The paper is generally well structured and well written. Nevertheless, there are some changes I would suggest in order to increase its scientific clarity:

- Please add a null hypothesis to the aim of your study

Thank you for this advice, which helps us to improve the quality of our manuscript. The null hypothesis was added to the introduction. 

“The null hypothesis of the present study was that the use of the novel polyurethane adhesive is equally effective and shows no differences in bone resorption between the use of the gold standard COSP after tooth removal under rivaroxaban therapy.”

- Were the extractions performed by the same researcher?

Thank you for your comment. We have added the following sentence to the “Materials and Methods” section: 

“All operations were executed by a single and experienced surgeon.”

- Did any extractions resulted in complications, such as alveolar bone or root fracture during the extraction?

Thank you for the thoughtful correction. The extractions were performed gently in accordance with existing protocols, so that no cases of bone fractures occurred. The few cases of root fractures were resolved without any difficulties, so that no roots of the first upper molar remained in situ. To clarify, we have added the following sentence to the manuscript:

“During the extractions no bone fractures occurred and in the rare case of root fractures these could be removed without any complications.“

- Why wasn't a control group, no COSP or VIVO and no anticoagulant, included in the study's design?

Thank you for this important objection. Originally, the authors planned a control group when approving the animal experiment project. However, in the course of the process of applying for approval of the animal experiment application, it was requested that we reduce the number of experimental animals by not using a control group in accordance with the 3R principles. Furthermore, it was argued that the healing of the alveolar bone after tooth extraction was sufficiently described based on the large amount of scientific literature with already published extractions in rats. The authors regret this fact and politely refer to the circumstances mentioned.

- The limitations of the study should be more broadly discussed

We would like to thank you for the opportunity to describe and discuss more broadly the limitations. Accordingly, we have added the following sentences to the discussion section:

“Nevertheless, the used rat model must be considered a limitation, as biological discrepancies in orofacial bone regeneration, including the healing of tooth extraction sockets, between rodents and humans are described in the literature [30]. Considering that no animal model perfectly matches the human situation, this limitation in the transferability of the observed results to humans must be critically considered.“

- No commercial interests to the COSP or VIVO should be disclosed

Thank you very much for this valuable advice. The sentence “The authors have no commercial interests in COSP or VIVO” was added to Conflicts of Interest

Kind regards,

Marius Heitzer and colleagues
